# Dietary Fiber from Underutilized Plant Resources—A Positive Approach for Valorization of Fruit and Vegetable Wastes

**Shehzad Hussain [1], Ivi Jõudu [1,2] and Rajeev Bhat [1,\*]** 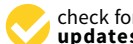

[1]   ERA Chair for Food(By-) Products Valorisation Technologies (VALORTECH), Estonian University of Life Sciences, Fr.R. Kreutzwaldi 56/5, 51006 Tartu, Estonia; Shehzad.Hussain@emu.ee (S.H.); ivi.joudu@emu.ee (I.J.)

[2]   Chair of Food Science and Technology, Institute of Veterinary Medicine and Animal Sciences, Estonian University of Life Science, Fr.R.Kreutzwaldi 56/5, 51006 Tartu, Estonia

\*   Correspondence: rajeev.bhat@emu.ee or rajeevbhat1304@gmail.com; Tel.: +37-2731-3927

**Abstract:** Agri-food industries generate enormous amounts of fruit and vegetable processing wastes, which opens up an important research area aimed towards minimizing and managing them efficiently to support zero wastes and/or circular economy concept. These wastes remain underutilized owing to a lack of appropriate processing technologies vital for their efficient valorization, especially for recovery of health beneficial bioactives like dietary fibers. Dietary fiber finds wide applications in food and pharmaceutical industries and holds high promise as a potential food additive and/or as a functional food ingredient to meet the techno-functional purposes important for developing health-promoting value-added products. Based on this, the present review has been designed to support 'zero waste' and 'waste to wealth' concepts. In addition, the focus revolves around providing updated information on various sustainability challenges incurred towards valorization of fruit and vegetable wastes for extraction of health promoting dietary fibers.

**Keywords:** dietary fiber; agri-food wastes; by-products; valorization; health benefits

---

## 1. Introduction

Higher production rate coupled with inappropriate handling technologies have led to the generation of enormous amounts of wastes in the food industries, particularly those from fruit and vegetable processing. This has opened up an important research area aimed towards minimizing and managing industrial wastes more efficiently to support zero wastes concept. Besides, food loss and waste reduction/management remain integral parts of the circular economy. As per the Food and Agriculture Organization (FAO) of the United Nations, annually about one-third of the global food production (~1.3 billion metric tons) is wasted [1]. Nevertheless, it is estimated that nearly half of the horticultural produce (fruits, vegetables, and root crops) is wasted globally, reaching up to 60% [2–4]. Earlier, Monier et al. [5] have quantified the amount of food wastes generated in the European Union, estimated to be ~180 kg of annual food loss per person. Further, recent estimation of food wastes by European Commission has estimated ~160 kg of food to be wasted per person [6]. According to Kader [7], around one-third of all fruits and vegetables produced globally is lost during postharvest process. Additionally, as much as 50% of the cultivated fruits and vegetables are wasted even before reaching the consumption stage [8,9].

Fruit and vegetable wastes generated in food industries post processing remain as underutilized owing to lack of appropriate processing technologies essential for their efficient valorization [8]. These vegetal wastes and/or by-products generated are a well-established source of bioactive compounds

and include health beneficial dietary fibers. The health promoting potential of dietary fiber includes lowering of blood cholesterol and sugar levels, improving cardiovascular health, and much more [9]. Nevertheless, dietary fiber holds high promise as a potential food additive or as a functional food ingredient, which can meet the techno-functional purposes required for developing health-promoting value added products [9]. In this view, as per the authors' knowledge, there is no single review providing in-depth information and discussing on the dietary fiber obtained from fruit and vegetable wastes. In the present-day scenario, sustainable utilization and management of food industrial wastes and/or by-products is vital to minimize pollution created by landfills. Keeping this as the background, the main aim of this review was to introduce novel concepts for effective reuse, recyclability, and maximal utilization of wastes and by-products for value addition as well as to boost the economic value. This review also supports 'zero waste' and 'waste to wealth' concepts, with the main focus relying on providing updated information on various sustainability challenges incurred towards utilizing fruit and vegetable wastes for extraction of dietary fiber, which is envisaged to find potential applications in various food industries.

## 2. Agri-Food Waste Valorization

Valorization technology involves sustainable conversion of agri-food wastes to value-added products. In the majority of the instances, agri-food wastes remain underutilized and find potential application only as bio-compost or as bio-fuel. Nevertheless, if left untreated for a long period, they can pose serious environmental stress, producing foul smells and pollution.

As per the available reports, agri-food wastes can be a good source of functional bioactive compounds [10]. Most of these bioactive compounds are proved to possess health beneficial properties such as antioxidant, antiviral, antibacterial, cardio-protective, anti-tumor, anti-obesity, etc. [11,12]. Because of post-processing extraction of the pulp (to produce juice, jams, purees), high amounts of wastes are generated [13].

Food processing industrial wastes include bioactives such as dietary fibers, pigments, essential minerals, fatty acids, antioxidant polyphenolic compounds, etc., and require green approaches to obtain these value added compounds [14]. Alkozai et al. [15] used powder obtained from pomace of pineapple, carrot, banana, orange peels, and mango kernel in preparation of cookies. Accordingly, cookies prepared with 10% of powder recorded higher acceptability scores on sensory evaluation. Reports are also available wherein apple pomace, apple fiber powder, apple skin, carrot pomace, orange pomace, and mango peels have been used to extract dietary fibers and used as a functional food ingredient in various bakery based products (cookies, crackers, cakes, muffins, biscuits, bun, wheat rolls, etc.) [16].

Principally, dietary fibers are carbohydrate polymers such as cellulose, hemicellulose, lignin and pectin, which provide structural rigidity to the plant cell wall. Depending on the water solubility, dietary fibers are categorized as soluble dietary fiber (SDF) and insoluble dietary fiber (IDF). In Figure 1, details on different types and sources of agri-food based dietary fiber are depicted. Accordingly, dietary soluble fiber types include pectin (sugars from whole grain, legumes, etc.), gums (sugar monomers from beans, legumes, etc.), and mucilage (aquatic plants, cactus, aloe vera, okra, as well as glycoproteins from food additives). Whereas, insoluble dietary fiber types include cellulose (providing glucose monomers obtained from fruits, root vegetables, etc.), hemicellulose (complex sugars from cereal bran and grains), and lignin (aromatic alcohols from vegetables). With regard to extraction methods identified, dietary fiber fibers (soluble and/or insoluble) are obtained via dry and/or wet processing, chemical methods, enzymatic gravimetric methods, and microbial methods (with certain limitations). More recently, green extraction methods such as water extraction, ethanol extraction and steam extraction, pulsed electric field assisted extraction, ultrasonic assisted extraction, high hydrostatic pressure assisted extraction, and other combination techniques are also widely used for extraction purposes [17,18].

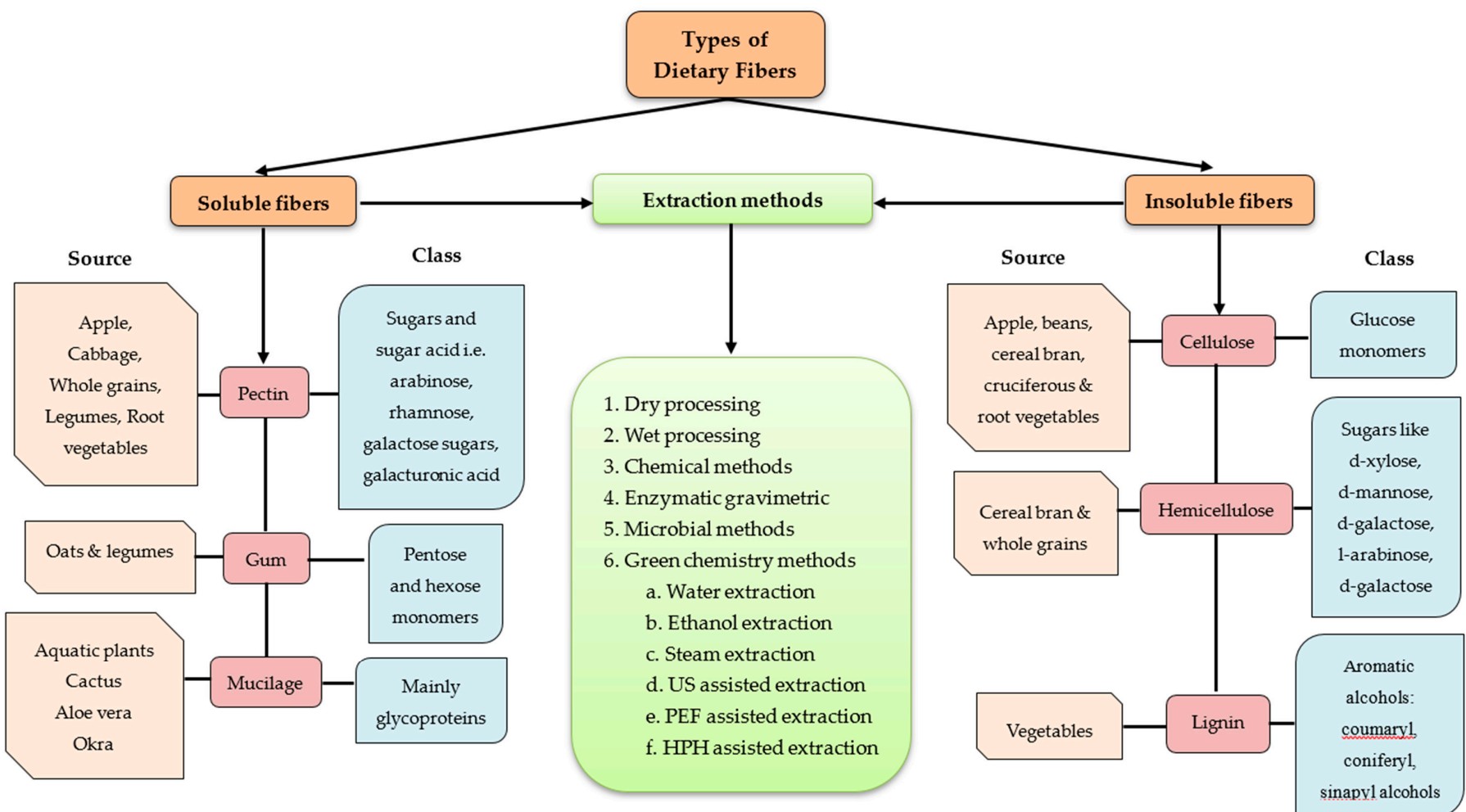

**Figure 1.** Agri-food based dietary fiber types, sources, and methods for their extraction.

To date, several techniques are recommended for dietary fibers extraction from plant resources. The techniques used for extraction like drying method, solvents extraction, and intensity of treatments were found to influence the composition and traits of obtained fibers [19]. Besides, selection of fiber extraction method can depend on the chemical nature of fiber, composition of particular fiber, presence of oligosaccharides, degree of polymerization, complexity, etc. [20]. Extraction method can also affect the behavior of dietary fiber in food applications as well as inside the human body [21]. Liquid to solid ratio, contact time, temperature, and selection of extraction method are some other factors that can influence the yield of dietary fiber [22]. Various treatment methods are reported to impart diverse effects on the structure of dietary fibers. Use of alkali and acid mediated extraction can damage the molecular structure of dietary fiber, however, enzymatic assisted extraction techniques can lead to incomplete extraction [17,23]. Alternatively, combined extraction methods (enzymatic and solvent) can also be useful for dietary fiber extraction. Further, modified wet-milling method has been recommended for better extractability, as these are cost-effective, capable of producing high purity fiber, and utilize minimal amounts of chemical and water than other routinely used methods. Purity of dietary fiber obtained by application of wet-milling technique is reported to be in the range of 50–90% [17].

Further, modern day processing techniques such as pulsed electric field, ultrasonic, microwave, high hydrostatic pressure, ionizing radiations, etc., are reported to have certain advantages and drawbacks. The use of these innovative, sustainable, and green extraction technologies supports high quality extraction that is reproducible and easy to handle, with lower environmental impact [24,25]. Sun et al. [26] used ultrasonic assisted alkali extraction (as combination technique) for extraction of insoluble dietary from soybean residues. The results indicated yield of insoluble fiber to be 744 mg/g of raw soybean. Recently, Wen et al. [27] have investigated on the impact of ultrasonic-microwave assisted extraction on soluble dietary fiber from coffee wastes (silver skin of the seeds) and compared it with conventional solvent extraction technique. In their study, the highest recovery rate (42.7%) of soluble dietary fiber was achieved by ultrasonic-microwave extraction (as a combination extraction technique). This was 1.9, 1.5, and 1.2-fold of the recovery rates achieved by microwave assisted, ultrasonic assisted, and conventional solvent assisted extractions, respectively. Begum and Deka [28] extracted dietary fiber form culinary banana bract by using ultrasonic-assisted extraction. The yield of soluble, insoluble, and total dietary fibers was 4.65, 78.7, and 83.9 g/100 g, respectively. In a similar study, hemicellulose and phenolic compounds were extracted from bamboo 'bast fiber' powder by application of ultrasonic assisted extraction technique. Analysis revealed that a combination of ultrasonic extraction and hot water treatment lead to an increase in the extraction efficiency to 2.6-fold. In addition, it also contributed to higher amounts of polyphenolic compounds, hemicellulose, and molecule of lignin biosynthesis [29]. Further, three different combination techniques, i.e., microwave-ultrasonic, microwave-sodium hydroxide, and microwave-enzymatic treatments, were compared for the extraction of soluble dietary fiber from peels of grapefruit. Results of this study revealed the yield of soluble, insoluble, and total fibers for microwave assisted extraction to be 7.9, 55.8, and 63.4 g/100 g, and for microwave-sodium hydroxide treatment it was 17.2, 46.2, and 63.9 g/100 g. For microwave-enzymatic treatment, the values were 9.2, 53.2, and 63.4 g/100 g; and for microwave-ultrasonic treatment it was 8.6, 55, and 63.8 g/100 g [30]. In another study, pulsed electric field technique was applied for extraction of cellulose from 'Mendong' fiber. Pulsed electric field assisted extracted cellulose showed better crystalline index when compared to alkali extraction process. This process increased the crystallinity of extracted cellulose by 83–86% [31].

However, these modern-day extraction techniques have certain drawbacks. For example: The main disadvantage can include high-energy consumption (as in microwaves), separation issues (as in ultrasonic extraction), and lack of user friendliness (as in pulsed electric field extraction technique) [25]. Hence, from our point of view, wet milling can be considered as an ideal extraction method due to its affordability and ability to produce high quality pure fiber.

Currently, on a global platform, food-processing industries are continuously exploring novel avenues to obtain dietary fiber from underexplored plant resources, to be used as a value-added

healthy ingredient. Generally, dietary fibers are obtained from cereals and/or their by-products. Garcia-Amezquita et al. [32] have reviewed on various aspects relevant to processing of plant by-products to obtain fiber-rich concentrate. Besides, various aspects relevant to evaluation of functional properties and technological functionalities of selected fruit and vegetable by-products are covered. Overall, in their article, the authors have detailed more on dietary fiber concentrates from fruits and vegetable by-products, whereas, in this article, we have covered all of the available information in more than 45 fruits and vegetables not only for SDF, IDF, and TDF, but also for pectin, cellulose, hemicellulose, and lignin. In addition, the main focus of the present review is to provide updated information on valorization opportunities of wastes/by-products derived from processing of various types of fruits and vegetables in a global context, the role of dietary fiber in health management, and finally the contribution to circular economy.

*Utilization of Fruit and Vegetable Wastes and By-Products on Circular Economy*

Sustainable management of fruit and vegetable wastes and/or by-products at the industrial level is important to minimize high volumes accumulated as landfills. In this regard, developing novel approaches for their effective reuse and recyclability can help in successful valorization and can contribute indirectly to boost the economic value. As these wastes and by-products can be a potential source of valuable bioactive compounds, including those of dietary fibers exhibiting a wide range of bioactivities, tapping their full potential will definitely contribute positively to the circular economy. Nevertheless, Europe alone generates around 88 metric tons of organic agriculture-based wastes [3], which are mostly discarded as landfills. Considering this scenario, effective management of wastes and/or by-products can be considered as a positive move from a linear economy to a circular economy [20,21]. Additionally, application of novel approaches are important to improve upstream recovery process of wastes leading to production of value-added compounds, which are all based on a sustainable circular economy concept [33]. Biotechnological methods and green processing technologies can be applied for effective extraction as well as to find future industrial applications for these wastes and by-products. In the perception of a circular economy, valorization of fruit and vegetable waste allows materials to be recycled or reused and placed back into the supply chain, thus allowing the economic growth as well as contributing for minimal negative environmental effects [34]. Besides, enormous amounts of wastes and by-products generated in the agri-food industries can lead to production of low-cost value-added products such as fragments of bio-refineries, fuel, valued added enzymes, flavoring components, natural pigments, health promoting components such as dietary fibers, vitamins, flavonoids, as a source of prebiotics, etc. In addition, they can also be essential components to produce functional foods [14,35–37]. All these contribute to achieving success on waste minimization, effective utilization, and zero wastes concept. Taste the waste is another notion, where bioactives recovered from wastes and by-products can be used as a value-added functional food.

## 3. Role of Dietary Fiber in Health Management

Dietary fiber is perceived to be an integral and compulsory part of a healthy human diet that can lower the risk of several diseases, including diabetes and cardiovascular diseases [38,39]. Besides, high intake of dietary fiber is linked with decreased risk of hypertension [40], obesity [41,42], diabetes [43], coronary heart disease [44], stroke [45], and certain gastrointestinal disorders [46,47]. Additionally, increase in consumption of dietary fiber is established to improve serum lipids profile [48,49], blood glucose [50], while it promotes regularity [51], lowers blood pressure [52,53], and assists in weight loss [54,55], as well as appears to improve immune function [56]. Potential applications and health benefits imparted by dietary fibers obtained from fruits and vegetables and their wastes are depicted in Figure 2.

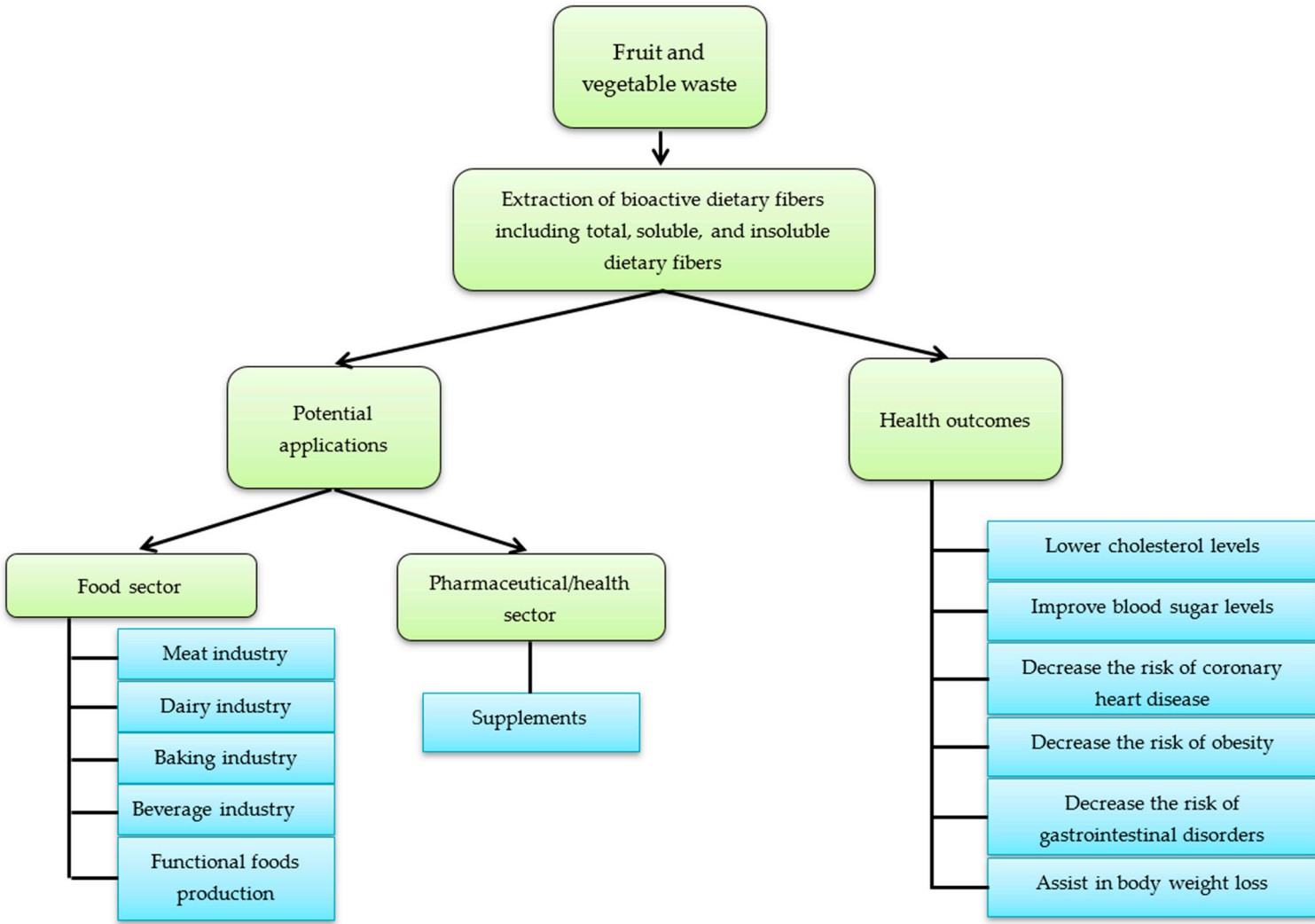

**Figure 2.** Potential applications and health benefits imparted by dietary fibers obtained from fruit and vegetable waste.

The Food and Drug Administration (FDA) of the United States Department of Health and Human Services has accepted two major health related claims for dietary fibers: The first claim affirms that a reduced intake of fat and high consumption of dietary fiber (derived from various sources such as vegetables, fruits, and/or grains) may decrease several kinds of cancer [57]. Recent research findings [58] support this view, linking increased consumption of dietary fiber with management of numerous kinds of cancers such as oral, colorectal, larynx, prostate, breast, and small intestine cancers [59,60]. A number of modes of action has been anticipated for the potential health benefits imparted. First, dietary fiber are not readily digested in the small intestine, thus letting it pass through the large intestine wherein the fermentation process takes place, leading to generation of short chain fatty acids (SCFA's), which possess anti-carcinogenic attributes [61]. Second, as dietary fiber enhances viscosity and fecal bulking, there is little contact time among mucosal cells and potential carcinogens to interact [62]. Third, dietary fibers enhance the binding among carcinogens and bile acids [63]. Fourth, elevated consumption of dietary fibers produces higher levels of antioxidants, and fifth, dietary fibers may enhance the quantity of estrogen excretion in the feces because of a reticence of estrogen absorption in the intestines [64,65].

The second FDA claim regarding the health benefits of dietary fibers affirms that diets low in cholesterol and saturated fat with high level of whole grain, fruits, and vegetables exhibit reduced threat from cardiovascular diseases [57]. For most of the world's population, a higher intake of dietary fiber is supposed to be approximately 25–35 g per day, of which 6 g is soluble dietary fiber (SDF). In fact, a number of research findings support the association between regular dietary fiber consumption with reduced risks of developing cardiovascular diseases. Though, more recent research establishes remarkable data demonstrating that for each 10 g of extra dietary fiber included in a diet, the death caused by coronary heart disease decreased by 17 to 35% [66]. Most common risk factors for cardiovascular disease include hypercholesterolemia, hyperlipidemia, obesity, hypertension, and diabetes (Type 2) [67,68].

Plant based dietary fiber constituents such as polyphenolics, tannin, lignin, inulin, and beta-glucan demonstrate several health beneficial effects [55,69]. Besides, certain types of soluble dietary fiber can serve as a potential prebiotic source and help in maintaining the balance of beneficial gut microbes along with imparting several positive health effects in humans. In addition, from all of the available reports in the scientific database, it can be established that regular consumption of these dietary fiber rich plant sources can impart potential health benefits to humans.

Dietary fiber extracted from vegetal wastes and by-products can be a rich source of polyphenols, flavonoids, and carotenoids (e.g., hydrolyzable polyphenols, hydrolyzable tannins, proanthocyanidins), and can constitute as a single matrix referred to as antioxidant dietary fiber. These dietary fibers can exhibit physiological effects of fiber as well as antioxidant properties in a single matrix. However, antioxidant capacity of dietary fibers can be negligible.

On the other note, reports are available wherein industrial food processing tends to increase health-promoting compounds along with dietary fiber. For example: Bender et al. [70] reported on the effects of micronization on phenolic compounds and antioxidant capacity of grape pomace and its dietary fiber concentrate. Accordingly, micronization significantly enhanced soluble dietary fiber and improved extraction of phenolic compounds, and increased antioxidant capacity. After micronization, total and insoluble dietary fiber decreased from ~75 to 67% and from 72 to 53%, respectively, while content of soluble dietary fiber increased from 4.4 to 15%. Phenolic compounds like gallic acid, flavan-3-ol catechin, epicatechin, epicatechin gallate, and total phenols increased from 0.26 to 0.61, 0.86 to 7.74, 0.6 to 3.6, 0.0 to 0.22 and (µg/mg d.w.), and 3053 to 5094 (mg GAE/100 g d.w.), respectively, in fiber concentrate. In a similar report, recently Chitrakar et al. [71] studied the effect of low temperature ball milling on bioactive dietary fiber powder from asparagus leaf by-product. Milling process increased chlorophyll content, flavonoids (150.4%), and total phenolics (26.1%) along with antioxidant activity (45.4%). Moreover, milling also increased soluble dietary fiber from 10.3 to 19.5%.

However, further research is warranted to identify the effect of novel green processing techniques on ex.

## 4. Fruit and Vegetable Wastes

Wastes encompass those which remain as unconsumed portions of fruits and vegetables (leaves, peel, skin, seed, skin, etc.), and/or those obtained as a consequence of morphological attributes, absence of appropriate handling processing technologies, or those which remain rejected for various reasons [72]. The amount and types of vegetal waste can differ from produce and vary according to the morphological parts, including roots, leaves, peel, pomace, stones, skin, pulp, and seeds [73]. The processing waste generated from fruit and vegetable industry is estimated to be around 25–30% [9,74]. Furthermore, the production of juice from fruits and vegetables yields about 5.5 MMT (Million Metric Tons) of wastes including its pomace [74–76]. Wine processing industries produce up to 5–9 MMT of solid waste from grapes and other fruits globally per year, which is composed of around 20–30% processed materials [74–76]. Other food processing industries such as those of frozen and canning industries contribute to nearly six MMT of solid waste yearly and this constitutes 20–30% of stalks, leaves, and stems [73].

For example, banana peel alone contributes to 35% wastage, while apples can generate up to 11% of pulp and seeds as waste (apple pomace), while industrial processing (mainly for juice production) of pineapples generates up to 30% wastage [77–80]. Mango processing yields 13% seeds, 11% peels, and about 18% of unusable pulp [75]. A report available has indicated stone and peel of mango to contribute up to 45% of total wastes [81]. Reports on grapes have indicated that seed, stem, and stalk contribute for 20% of the biomass as waste [77]. In citrus fruits, up to 50% of the wastes are from peel, skin, and seeds [9,77]. Further, with regard to vegetables, research works undertaken are much less quantitatively. However, most extensive studies on vegetables include peel wastes of potato [77], (15%), onion (17–38%) peel wastes [82], tomato peels (20%) [77], pea shell (40%), etc. [83]. However, several studies have demonstrated higher amounts of bioactive compounds and essential nutrients to be present in these plant wastes [84,85].

In the following sections, we have comprehensively covered some of the vital research reports relevant to dietary fiber from fruit and vegetable wastes.

## 5. Dietary Fiber from Fruit Processing Wastes

### 5.1. Apples

After juice extraction in the processing industries, apple pomace is usually discarded as a waste material. However, these wastes can be a good source of dietary fibers. Apple peel is reported to contain higher amounts of dietary fiber than pulp portion. Apple pomace had ~15% and 36% of soluble dietary and insoluble dietary fiber, respectively (d.w.) [86]. On the other note, Bae et al. [87] have reported 1.1% and 12% of soluble and insoluble dietary fiber in whole apple powder. It is studied that the fiber content from apple pomace can be recovered and utilized as a potential food ingredient [88–90]. Earlier, Yan and Kerr [91] reported total dietary fiber to vary between 44.2–49.5% in vacuum dried pomace, and this variation was non-significant when compared with freeze-dried pomace (48%). By using different extraction techniques, production of soluble dietary fiber can be enhanced. In particular, ultrasound-assisted extraction technique can provide higher yield of soluble fiber content as compared to microwave-assisted extraction or hydrolysis techniques [92].

Cellulose, pectin, hemicellulose, and lignin containing products is reported to have high water holding capacities between 9–10 g/g. Potential applications for these fiber containing products include bread and other baked stuff, dairy products, pharmaceuticals, as well as pet food [93,94]. In a study by Chen et al. [95] dried apple fiber was compared with oat and wheat bran in various bakery based products. Accordingly, a higher amount of total dietary fiber in apple was recorded when compared

to oat and wheat bran. Apple fiber composition showed 40% of cellulose and 19% of water-soluble hemicellulose (d.w.).

## 5.2. Berries

Berries are one of the best dietary sources that can be consumed raw or after processing into juice. Some of the popular and healthy berries include blackberries, blueberries, strawberries, raspberries, red, white, and black currants, sea buckthorn, and others. The wild varieties/cultivars of these also exist. In majority of the instances, the by-product obtained after juice pressing is referred to as berry pomace or the press cake and usually contains skin, stem, and seed parts. Dietary fiber occurs as pectin, lignin, cellulose, hemi-cellulose, and inulin [96,97].

In a study on berries by Reißner et al. [96], the pomace powder of blackcurrant, red currant, gooseberry, rowanberry, and chokeberry was reported to have high fiber content (>550 g/kg). In the study, soluble dietary fiber content was 3.9, 7, 7, 7.7, and 7% (d.w.), while the respective values for insoluble dietary fiber content was 5.6, 5.2, 4.9, 5.9, and 5.2% (d.w.) in blackcurrant, red currant, gooseberry, rowanberry, and chokeberry, respectively.

White et al. [98] have reported dried cranberry pomace to contain 66% and 6% (d.w.) of insoluble and soluble dietary fiber, respectively. Further, Wawer et al. [99] have also reported 72% of total, 66.6% of insoluble and 5.4% of soluble dietary fiber in chokeberry. In cranberry pomace, Gouw et al. [100] reported 58.7% of total, 58% of insoluble, and 0.8% (d.w.) of soluble dietary fiber, while in blueberry pomace, 50%, 49%, and ~1% (d.w.) were the total, insoluble, and soluble dietary fibers, respectively. In a similar study, Šarić et al. [101] evaluated fiber concentrate of blueberry and raspberry and recorded 53 and 56% of total dietary fiber, 50 and 55% of insoluble dietary fiber, and 2.1 and 1.7% (d.w.) of soluble fiber, respectively. Gouw et al. [100] have also evaluated the dietary fiber content in raspberry pomace and reported 38.4% of total, 38% of insoluble, and 0.3% of soluble dietary fiber. Dietary fiber in sea buckthorn and black currant was evaluated by Linderborg et al. [102], who reported this to be 21% and 24% (d.w.), respectively. Alba et al. [103] reported blackcurrant pomace from two different locations (in United Kingdom and Poland) to be 30 and 25% of soluble fiber, and 46.9 and 47.4% of insoluble fiber, respectively. Wawer et al. [99] reported 66.8% of total, 60% of insoluble, and 6.7% (d.w.) of soluble dietary fiber in blackcurrant. Similarly, Jakobsdotti et al. [104] reported 7.8% of soluble dietary fiber, 56% of insoluble dietary fiber, and 64% of total dietary fiber. In the same report, bilberry press cake was reported to have 5.8% of soluble dietary fiber, 58% of insoluble fiber, and 64% (d.w.) of total dietary fiber.

Dietary fibers obtained from various berries are mainly the carbohydrate-based polymers which includes: lignin, pectin, cellulose, and hemicellulose. Soluble dietary fiber was extracted from residue/waste of *Cornus officinalis* by using ultrasonic assisted extraction method, and actual yield was in the range of 12.24 + 12.89% [105]. Soluble dietary fiber was also extracted from bilberry, blackcurrant, and raspberry by hot water extraction, and yield was in ranged from 34–250 g/kg. Similarly, hot water extraction was applied to extract soluble dietary fiber cranberry pomace for 30 min at 75 °C with 1:5 solid to liquid ratio, and it was found that the extracts comprised of 887 g/kg of carbohydrates [103,106].

## 5.3. Grapes

Grape pomace is a potential source of dietary fiber and includes cellulose and hemicellulose with small amounts of pectic substances [107]. Vliente et al. [108] studied grape pomace as a possible dietary fiber source. In the grape pomace, total dietary fiber content recorded was 78% (d.w.), among which 9.5% was soluble dietary fiber and the remaining 68% was insoluble dietary fiber. Results of this study revealed that grape pomace can be efficiently utilized as a high dietary fiber additive in food products. Gonzalez-Centeno et al. [109] evaluated by-products of 10 different grape varieties. Accordingly, the results revealed that Tempranillo (red grape cultivar) had 36.9% of dietary fiber in the pomace, with 35% in the stem and 5% in fruit (expressed on f.w. basis). The total dietary fiber content (72%, d.w.) of red grape pomace was the same as with the pomace of white grapes. In another

study, soluble dietary fiber content (10%) (d.w.) was recorded to be less than insoluble dietary fiber (61% (d.w.) in the pomace of white grapes [110]. Deng et al. [111] investigated the pomace of various varieties of white and red dried grape skin and reported red wine grape pomace to contain total dietary fiber ranging between 51–56%, while the corresponding values for pomace of white wine grape ranged between 17–28%. In an another study, Pedras et al. [112] evaluated red wine grape pomace (obtained from Portuguese wine producer Esporão) for its bioactive compounds and found 15% cellulose and 11% hemicellulose (d.w.). Likewise, Bender et al. [70] evaluated the effects of micronization on the dietary fiber from grape pomace (obtained from red winemaking process) on the dietary fiber and recorded 66% of total dietary fiber encompassing 61% of insoluble and 4% of soluble dietary fiber (all expressed as d.w.). Oil and water holding capacities of seed fibers obtained from lemon, orange, and grape fruit were found ranging between 3.4–4.2 g/g and 4.8–7.8 g/g, respectively, with swelling capacities in the range of 22.4–24.6 mL/g [113].

*5.4. Mango*

Mango processing for extraction of pulp for obtaining juice and other value added products tends to generate 35–60% wastes, which can be a potential source of dietary fibers [114]. Mango peels and its fibrous pulp contains around 51% of total dietary fiber content [75,115]. Mango kernel and seed has 1.8 to 2% crude dietary fiber [116–118]. Compiled data on crude fiber contents in mango seed kernel is reported to range between 1.65–3.96% [119]. Ajila and Rao [120] evaluated total dietary fiber in mango peels and revealed their content to be 40–72%, with glucose, galactose, and arabinose as the main neutral sugars in soluble and insoluble dietary fibers. Ashoush and Gadallah [121] investigated mango kernel powder and mango peel powder and reported them to contain 0.3% and 9.3% (d.w.) of crude fiber, respectively. Moreover, "Tommy Atkins" (a variety of mango) had 28% of total dietary fibers including 14.3% soluble and 13.8% insoluble dietary fiber [122]. A product of dietary fiber made from mango peels residue after production of syrup was investigated by Larrauri et al. [123]. Wet milling and washing of the mango waste generated a fiber product which contained 7% polyphenols/kg fiber and 28.1% soluble fiber (d.w.). This product was also reported to possess a higher water holding capacity of 11.4 g/g fiber [123,124]. Moreover, dietary fiber extracted from mango peel had 11.4 g/g and 2.7 g/g of water and oil holding capacity, respectively [123]. Further, mango peels contain ~23% of soluble and 28 to 50% of insoluble dietary fiber with total sugar content in soluble dietary fiber ranging between 66 and 74% and insoluble dietary fiber between 73–82% [120].

*5.5. Orange*

Pulp and peels of orange wastes generated from orange juice extraction process comprised around 35–37% (d.w.) of total dietary fiber, which was rich in cellulose and hemicelluloses (range: 17–18%), pectic constituents up to 17%, and lignin between 2–3% (all on d.w.) [125,126]. Citrus fiber obtained from orange juice had 22% soluble and 54% insoluble dietary fibers. This fiber had a water holding capacity of 11:1, as well as 3–4 g/g oil binding capacity, that is 3 to 4 times of its weight [127]. The fiber obtained from citrus had low calories, elevated water holding, and oil absorption properties. Orange fiber is recommended to be a potential source of pulp for beverages (cloudy beverages), gelling agent, binder as a low caloric bulking agent, and as a thickener [128,129]. Studies have revealed that organoleptic properties of citrus dietary fiber do not adversely alter any of the food properties. Its recommended applications include dairy products, baby foods, drinks, soups, fruit juices, and desserts [127,130].

‘Liucheng’ variety orange peels contained 57% of total dietary fiber, of which 47.6% was insoluble dietary fiber fraction while 9% (d.w.) was soluble dietary fiber fraction. Cellulose and pectic polysaccharides were recognized to be their main components [131]. Total dietary fiber content in lemon peels was recorded to be 14% (d.w.), which is almost double than that of peeled lemon (7% by d.w.) [129,132]. Among the total dietary fibers, soluble and insoluble fiber fraction was 5% and 9% (d.w.), respectively. Pulp and peels of lemon contained considerable amounts of soluble and insoluble

dietary fibers. However, the amount of dietary fiber in the pulp was much higher (78%) than that of peel (53%) [132].

## 5.6. Peach

Peach has been reported to contain 30–36% of dietary fibers, of which soluble dietary fibers constitute around 12% and insoluble fibers are 24% (all on d.w. basis) [133]. Pulp and peels obtained as a by-product from peach juice extraction process had a dietary fiber content ranging between 31–36% (d.w.) with 20–24% insoluble dietary fiber as a major fraction. The soluble fiber fraction ranged between 9–12%, and this was higher than soluble dietary fraction found in cereals and grains [134].

Kurz et al. [135] found pectin to be the major form of polysaccharides in the cell wall of peach. Accordingly, peach fruit dietary fiber is recommended as a potential functional food ingredient. Suggested applications for use of peach dietary fiber as a functional ingredient include meats, bakery products, low caloric beverages, and extruded products [134].

## 5.7. Pear and Kiwi

Martin-Cabrejas et al. [136] evaluated dietary fiber products made from pomace remains (via membrane ultrafiltration) of kiwi and pear purees. It was found that pear and kiwi pomace contained 44% and 26% (d.w.) of total dietary fiber, respectively. Soluble fibers in both of the fruit pomaces were higher in methoxyl pectin. Nawirska et al. [137] studied the pear pomace for its dietary fiber fractions and found 13% pectin, 19% hemicelluloses, 39% cellulose, and 34% lignin.

Yan et al. [138] studied the effect of superfine grinding treatment and hydrostatic pressure on pear pomace. Results indicated that application of both treatments significantly increased soluble dietary fiber content from 10 to 16%, with a corresponding decrease in insoluble dietary fiber from 47 to 43%. Besides, an increase in the water holding and oil holding capacities from 3.4 to 5.8 g/g and from 1.8 to 2.8 g/g was recorded, respectively. In a study by Soquetta et al. [139], dietary fiber in the flour prepared from skin and bagasse of kiwi fruit had ~28% of total dietary fiber, 19% of insoluble dietary fiber, and 10% of soluble dietary fiber. Additionally, ~29% total dietary fiber, 24% insoluble dietary fiber, and 6% of soluble fiber were recorded in ripe bagasse flour of 'Monty' variety of kiwi fruit.

## 5.8. Pineapple

Dietary fiber obtained from shells of pineapple contained around 70.6% of total dietary fibers, of which the majority of fibers are insoluble. Glucose and xylose have been detected in fiber product as the main natural sugars [9,140]. Higher antioxidant activity of pineapple fiber has been reported, which was linked to polyphenol myricetin. The antioxidant activity was much higher than that of citrus and apple fiber. The dietary fiber obtained from pineapple is reported to possess natural flavor and color characteristics, which probably improve the acceptability of the product as a fiber supplement [9]. Earlier, Larrauri et al. [141] had investigated the preparation and formulations of a powdered drink supplemented with pineapple peel dietary fiber. Organoleptically acceptable high-quality pineapple peels with 8% of moisture content were crushed and ground (made into flour) and mixed with sugar, citric acid, flavor, color, and foaming agent to produce novel drink mix ('FIBRALAX'). This drink mix with ~25% of dietary fiber delivered laxative effects [142].

## 5.9. Pomegranate

Pomegranate peels/skin, an underestimated waste or by-product, is reported to provide a range of health benefits including those of hypolipidemic, anti-inflammatory, and antimicrobial (antiviral and antibacterial) activities [143,144]. Peels are also considered an eco-friendly waste due to its use as a reducing agent in making silver nanoparticles. Further, Kushwaha et al. [144] studied the fresh and detanninated peel of pomegranate for their nutritional value and reported detanninated peel to contain ~24% crude fiber, 29% neutral detergent fiber, 26% acid detergent fiber, and 8% of lignin. Fresh peel powder had ~13% crude fiber, 18% neutral detergent fiber, 15% acid detergent fiber, and 15%

lignin. In a similar study, Colantuono et al. [145] reported pomegranate peels-microsphere to have ~14% total, 12% insoluble, and 2% soluble dietary fiber, respectively.

### 5.10. Dietary Fiber in Exotic Fruits

Some of the exotic and popular fruits consumed for taste and health benefits include rambutan, jackfruit, durian, mangosteen, passion fruit, etc. In other popular fruits, the seeds and rind/skin portions contribute to high biomass production. For example: In durian, jackfruit, and mangosteen, the seeds and skin waste is up to 70% [83,146–148]; in passion fruits, the waste generated is up to 50% [149,150], in dragon fruit, it is 30–45%, in papaya, it is 10–20%, and in rambutan, between 50–65% [151,152]. Rambutan fruit peel is reported to contain dietary fiber of 0.9% [153]. Further, unripe and fully ripe durian fruit flour contained 8.20% and 10.14% of crude fiber, while in dehulled durian seed flour was revealed to contain: Approximately 5% crude fiber [154]. In jackfruit peel, Trilokesh and Uppuluri [155] reported 20% cellulose, 24% hemicellulose, and 2% of lignin (d.w.). In a similar study, Selvaraju and Bakar. [156] reported 54% cellulose, 23% hemicellulose, and 2.6% lignin (d.w.) in peel of jackfruit.

Winuprasith and Suphantharika [157] studied dried mangosteen rind and found total dietary fiber to be ~93% with insoluble dietary fiber being 78% and cellulose, 68% (d.w.). In peel and de-pectinised peel waste of passion fruit, Hernández-Santos et al. [158] reported ~58 and 72% of total dietary fiber with 46 and 65% being insoluble and 12% and 6.6% (all in d.w.) being soluble dietary fiber, respectively. In a similar study, in peel of passion fruit, 54% insoluble and 3% (d.w.) soluble dietary fiber is reported [159].

With the available databases, it seems research works undertaken on exotic fruits wastes are minimal and more works are warranted on this regarding extraction of dietary fiber and finding a potential application for them.

## 6. Dietary Fiber from Vegetable Processing Wastes

### 6.1. Carrot

Bao and Chang [160] studied the blanching effects on properties of carrot pulp waste remaining after extraction of juice. The range of total dietary fiber was 37–48% in blanched carrot pulp, reducing sugars 8–9%, protein 4–5%, and minerals 5–6% (d.w.). Blanched pulp of carrots also contained 31–35% of carotene with water retention capacity being 9–11%. Further, Chau et al. [161] studied carrot pomace for its dietary fiber content, which was 63%, containing 50% of insoluble and 14% of soluble dietary fiber (all in d.w.). They also isolated various fiber fractions from carrot pomace including insoluble fiber, water insoluble solids, and alcohol insoluble solids. Accordingly, carrot pomace had adequate amounts of insoluble fiber fractions, i.e., 50% of insoluble dietary fiber, 67% of alcohol insoluble solids, and 56% of water insoluble solids. Total dietary fiber was ~63%, and was composed of cellulose, hemicellulose, and polysaccharides. Further, fresh peels of carrots investigated for dietary fiber after blanching process indicated total dietary fiber content to have increased from 45 to 73% after blanching [162].

### 6.2. Cauliflower

Cauliflower is a nutrient rich vegetable with high waste index, owed to their non-edible portions consisting of leaves and stem [163]. These wastes can be good source of dietary fiber [164,165].

Femenia et al. [166] studied the properties of dietary fiber in dehydrated cauliflower and effects of its supplementation on sensory and physical properties of model foods. Each part of cauliflower had varied amounts of dietary fiber: Lower stem of cauliflower contained 65% dietary fiber, upper stem contained 48% of dietary fiber, and florets had 40% of dietary fiber (all in d.w.). Lower stem portion of cauliflower contained 51% cellulose and 21% hemicellulose, while upper stems and florets had 50–60% pectic substances [163,166].

Effects of various technological processing such as freezing, cooking, and blanching on dietary fiber, minerals, protein, and fat was evaluated in white and greenish yellow floret of cauliflower [167]. The study revealed that food processing methods can significantly affect the dietary fiber content and nutrients as well, and can be a key aspect regulating the level of nutrient loss. The cooking process of raw cauliflower resulted in minimal nutrient loss compared to cooking of frozen cauliflower [167–169]. Cauliflower leaves are reported to contain 25% cellulose, 8% hemicellulose, 19.45% acid detergent fiber, and 73% (all in d.w.) neutral detergent fiber [170].

### 6.3. Corn

Corn is a stable and nutritious food for humans and finds wide usage in various food and feed applications. Corn bran is a common by-product obtained after wet milling of the kernels [171], which can encompass ample amount of dietary fiber. Further, after harvesting, stems (corn straw) and whole ears of corn that remain as waste can be a good source of fiber. In a study conducted on ground corncobs by Anioła et al. [172] over a period of 4 different years, ground corncobs were reported to have total dietary fiber ranging between 89.9–93.2%, soluble dietary fiber in the range of 0.8–2.2%, Hemicellulose between 43–46%, lignin between 3.2–6.4%, and cellulose between 35–39% (all in d.w. basis).

Li et al. [173], working on the nutritional composition of corn stover, reported significant variations in the fiber contents. This study was aimed towards finding a potential application for corn stover as a ruminant feed. Results showed ear husk, leaf blade, and stem pith to possess high nutritive value when compared to some of the other corn stover fractions. Accordingly, the leaf blade had lowest neutral detergent fiber and acid detergent fiber content of 62 and 31%, respectively. While, the stem rind had high acid detergent fiber and acid detergent lignin of ~48% and 8%, respectively. Further, the ear husk exhibited the neutral detergent fiber content to be ~83% and relatively lower acid detergent lignin of 3.6%.

### 6.4. Onion

All layers of onion contain ample amounts of dietary fibers in various proportions. Jaime et al. [174] evaluated three different varieties of onion (skin and inner layers) for dietary fiber content. Accordingly, ~63% (d.w.) of total dietary fiber was found in skin, which was the highest when compared to all other parts of "Grano de Oro" onion, while the lowest total dietary fiber content was detected in the inner part (12%, d.w.). Besides, higher amounts of insoluble dietary fibers (67%, d.w.) were also found in the skin of "Grano de Oro" variety of onion compared to that in the inner part.

By means of sterilization and pasteurization, Benitez et al. [175] stabilized triturated onion wastes, liquid fractions, and solid leftover (residues). From the study, industrial processing was reported to impart a significant influence on composition of bioactive compounds. Pasteurization was the most appropriate treatment to ensure the safety of dietary fiber-enriched products. This investigation also revealed bagasse to contain high levels of dietary fiber, which ranged from 36-45% (d.w.).

### 6.5. Potato

Potato wastes, mainly those of peel, encompass high amounts of bioactive compounds, including those of dietary fiber [176]. Effects of thermal processing on potato peel dietary fiber composition and its hydration properties have been identified [177]. Extrusion cooking and baking resulted in elevated total non-starch polysaccharides content in peels of potato. However, only extrusion cooking was related to an elevated level of soluble to insoluble ratio of non-starch polysaccharides. Processing led to a reduction in the content of 'Klason' lignin; however, this did not affect the content of uronic acids. Both extrusion and baking resulted in a decline in the water absorption capacity of potato peels [178,179].

Variations in the composition of dietary fibers obtained from potato peels subjected to extrusion cooking and peeling has been investigated. Extrusion cooking was associated with increased lignin

and total dietary fiber content with decreased starch content in steamed potato peels. Although lignin content decreased, total dietary fiber remained unaffected during extrusion of peels. As a result of extrusion cooking, an increase in soluble non-starch polysaccharide in both types of peels was observed [177,180].

Five polish potato varieties, namely Raja, Saturana, Courage, Rosalind, and Hermes were investigated for dietary fiber content. Raja had 10.3%, Saturana had 9.9% (d.w.), Courage had 9.3% (d.w.), Rosalind had 9.7% (d.w.), and Hermes had 9.5% (d.w.) of dietary fiber [181]. Ncobela et al. [182] found peels of potato to contain approximately 6.1–12.5% (d.w.) crude fiber. Solid wastes of potato have also been reported to be a rich source of dietary fiber, the range of which was between 27–35% (d.w.) [164,183]. Overall, potato pulp can be considered as a good source of dietary fiber that remains as an underutilized waste or by-product of the potato starch processing industries [184].

### 6.6. Tomato

Tomato pomace contains around 50% (d.w.) dietary fibers [185]. In one of the studies by Herrera et al. [186], dietary fiber extracted from dried tomato peels had 83% of total dietary fiber with 10:1 ratio of soluble to insoluble dietary fibers. The tomato processing industry produces considerable amounts of fresh tomato wastes, and this can be preserved via silage [187]. Tadeu Pontes et al. [188] recorded a two-fold increase in yield of soluble dietary fiber (up to 108%) by extruding (via use of single-screw extruder) tomato pomace with corn semula and a starch component. Tomato pomace was also evaluated for its soluble fiber fractions as well as acid and neutral detergent fibers and 9–12% hemicellulose, 12% cellulose, 37–44% neutral detergent, and 46–51% (all d.w.) acid detergent fiber were recorded [189].

Details on dietary fiber content (total, soluble and insoluble, pectin, cellulose, hemicellulose, and lignin) in the processing wastes of selected popular and commonly consumed fruits and vegetables is provided in Table 1. Overall, from the available literature, it is evident that research works undertaken on dietary fiber extraction from wastes, characterization, and potential applications remain in infancy levels. Hence, there is an enormous scope and opportunity to explore the wastes, including those of underutilized fruits and vegetables for its potential applications.

### 6.7. Underexplored Vegetable Wastes and Grain By-Products

Underexplored vegetable wastes include those produced by pumpkin, mushroom, celery, crucifer members, lotus root, etc. Cerniauskiene et al. [190] evaluated pumpkin cultivars including *Cucurbita pepo* (*Herakles, Golosemiannaja, Miranda, Danaja* and *Olga*) and *Cucurbita maxima* (*Chutorianka, Zalataja grusha, Arina, Chudo judo, Kroshka*) for the dietary fiber content, and reported *Kroshka* cultivar to have high neutral (26.5%) and acid dietary fibers (24.6%), while *Chutorianka* was rich in water soluble carbohydrates (61.2%). On the other hand, *Herakles* cultivar was found to be high in natural dietary fiber (26.5%) and acid dietary fiber (23.5%) (All values in d.w.).

Consumption of wild mushrooms is increasing due to their rich bioactive compound profile. Eleven mushroom species were evaluated by Cheung [191] for their dietary fiber content. Accordingly, it was reported that total dietary fiber ranged between 27 and 44%, with insoluble dietary fiber ranging between 25 and 41% and soluble dietary fiber between 0.6 to 2.8% (d.w.). Further, in a study by Nile and Park [192], 20 wild mushroom species were evaluated for dietary fiber content. They reported total dietary fiber to range between 27 and 36%, insoluble dietary fiber between 12 and 21%, and soluble dietary fiber between 2 to 4%.

Crunchy stalks of celery make it a famous low caloric vegetable with a range of health benefits. According to Sowbhagya et al. [193] celery seed contained 56% total dietary fiber, 49% insoluble dietary fiber, and 7% soluble dietary fiber, while celery spent residue (after oleoresin and oil extraction) had total dietary fiber of 61%, insoluble dietary fiber of 53.5% and soluble dietary fiber of 7.5% (d.w).

Due to its distinctive taste, lotus seeds and root are widely consumed in many of the Asian and African countries. Lotus root processing generates a high volume of waste from the non-edible parts,

especially those of nodes. However, nodes of lotus root can be a good source of dietary fiber [194]. Further, Hussain et al. [195] studied the effect of various micronization treatments on fiber rich fractions of lotus nodes and found total dietary fiber ranging between 39.6–41.5%, insoluble dietary fiber from 29.9–37.8%, and soluble dietary fiber from 2.8–13.4% (d.w.). Besides, lotus seeds are reported to have crude fiber of 2.7% [196]. Sugar beet leaves, radish leaves, pea vine, and cabbage leaves have been reported for cellulose and hemicellulose contents. According to Wadhwa and Bakshi [170], sugar beet leaves contain 11.4% cellulose and 21.2% hemicellulose; radish leaves contain 14.9% cellulose and 5.9% hemicellulose; pea vines have 36.8% cellulose and 10% hemicellulose; and cabbage leaves have 13.7% cellulose and 11.1% (all d.w.) hemicellulose.

**Table 1.** Content of dietary fiber reported in selected fruits and vegetables processing wastes (%) (d.w.).

| Waste Type | Produce | SDF | IDF | TDF | Pectin | CE | HC | LI | References |
|---|---|---|---|---|---|---|---|---|---|
| *Fruits* | | | | | | | | | |
| Whole | Apple | 22 | 63 | 86 | 6–8 | - | - | - | [197,198] |
| Pomace | Apple | 19 | 70 | 89 | 7–23 | 44 | 24 | 20 | [197,199] |
| Pomace | Raspberry | 3 | 75 | 78 | - | - | - | - | [200] |
| Pomace | Cranberry | 5 | 53 | 58 | 11 | 74 | 26 | 43 | [201,202] |
| Pomace | Chokeberry | 5 | 70 | 75 | - | - | - | - | [201] |
| Pomace, stalk | Grapes | 11 | 64 | 74 | 32 | 38 | 14 | 33 | [203] |
| Stalk | Grapes | - | - | - | - | 30.3 | 21 | - | [204] |
| Pomace skin | Wine grapes | - | 16–52 | 17–53 | - | - | - | - | [111] |
| Whole | Mango | 28 | 41 | 70 | - | 27 | 54 | 19 | [205] |
| Peel | Mango | 19 | 32 | 51 | 18–32 | - | - | - | [115,120,206] |
| Peel | Orange | 9–22 | 41–48 | 57–63 | - | 14 | 6 | 2 | [131,164] |
| Pomace/pit | Peach | 19 | 36 | 54 | - | 31 | 22 | 27 | [198,207] |
| Pomace | Pear | 7–10 | 28–46 | 35 | 13 | 35 | 19 | 34 | [136,138] |
| Pomace/Bagasse | Kiwifruit | 5–7 | 13–23 | 2–30 | - | - | - | - | [136,139] |
| Skin | Kiwifruit | 9.4 | 18.7 | 28.2 | - | - | - | - | [139] |
| Leaves, stem | Pineapple | 0.6 | 75 | 76 | | 30–42 | 32–37 | 19–22 | [208] |
| Peel | Pomegranate | 13 | | | | 33 | 27 | 15–28 | [145,209] |
| Peel/waste | Banana | - | - | 50 | - | 26 | 20 | 14 | [210] |
| Flesh | Date | 5–7 | 9–11 | 14–18 | 2.7 | 24 | 27 | 22 | [211–213] |
| Peel | Grapefruit | 4–17 | 46–60 | 62–63 | 16–25 | - | - | - | [30,214,215] |
| Waste | Jackfruit | - | - | - | - | 8 | 23 | - | [216] |
| Seeds | Jackfruit | - | - | - | - | 18.8 | 16.2 | - | [216] |
| *Vegetables* | | | | | | | | | |
| Whole | Carrot | - | - | - | 9–10 | - | - | - | [198] |
| Pomace | Carrot | 13–24 | 45–50 | 64–70 | 3.9 | 52 | 12 | 32 | [131] |
| Whole | Cauliflower | - | - | 16.2 | - | - | - | - | [217] |
| Waste | Cauliflower | - | | 35 | - | 17–19 | 14–15 | 8–11 | [218,219] |
| Corncobs | Corn | 0.8–2 | - | 90–93 | | 35–39 | 43–46 | 3–6 | [172] |
| Skin/leaves | Onion | - | - | 68.3 | - | 41 | 16 | 39 | [174,220] |
| Peel | Potato | 10–20 | 20–53 | 73 | - | - | - | - | [221,222] |
| Pulp/whole | Potato | - | - | - | 10–12 | 4 | 14 | 0.4 | [198] |
| Whole | Tomato | 8.9 | 40.5 | 49.5 | 6.4–6.9 | - | - | - | [198,223] |
| Pomace | Tomato | - | - | 59 | - | 9 | 5 | 3 | [94,185] |
| Pomace/waste | Pumpkin | - | - | 77 | - | 11 | 6.4 | | [216,224] |
| Skin | Pumpkin | - | - | - | 25 | - | - | - | [225] |
| Whole | Mushroom | - | 27–44 | 25–41 | - | - | - | - | [191,192] |
| Husk | Garlic | 4.2 | 58.1 | 62.2 | - | 42 | 21 | 35 | [220,226] |
| Hulls | Pea | 4.1 | 87.4 | 91.5 | - | 17 | 33 | 2.5 | [227] |

SDF, soluble dietary fiber; IDF, insoluble dietary fiber; TDF, total dietary fiber; CE, cellulose; HC, hemicellulose; LI, lignin.

Still, there are wide gaps on information generated in relevance to dietary fiber and health effects of some popular and underutilized vegetables and their wastes (e.g., beetroot, tubers, turnip, pumpkin, wild vegetables).

## 7. Bioactive Compound and Dietary Fiber in Fruit Seeds of Industrial By-Products

Fruits and vegetable wastes and by-products such as pomace and press cake have been extensively studied for bioactive compounds including dietary fibers. Several fruit seeds obtained from industrial wastes or as by-products have also been studied for bioactive compounds, however, only few studies are available on fruit seeds. Górnaś et al. [228] studied the composition of phenolic compounds in flesh, seeds, and stems recovered from crab apple pomace and highest content of polyphenols (3592.6–23,606.8 mg/kg) was detected in seeds. Further, Górnaś et al. [229] studied the lipophilic composition of eleven apple seed oils obtained from industrial by-products generated during juice processing and fruit salad preparation. In this study, the oil yield in apple seeds ranging from 12–27 g/100 g with palmitic, oleic, and linoleic acids to be the dominant fatty acids. Seed oils obtained from industrial fruit waste and by-products are considered to be a rich source of tocotrienols and tocopherols [230]. Oils recovered from the seeds of industrial fruit wastes and by-products from apples, gooseberries, grapes, pomegranate, watermelon, Japanese quince, red current, sea buckthorn, and honeydew were investigated for phytosterols and fatty acid composition. Significant amounts of oil were recovered with highest yield of 28.5% in watermelon seeds, and with β-sitosterol as the major phytosterol in all of the nine fruit seed oils [231]. In an another study, Górnaś [232] investigated tocochromanol profile in seed oils of industrial by-products of seven crab and five dessert apple varieties grown in Latvia. This study revealed seed oils recovered from dessert apples (191–380 mg/100 g oil) to be high in tocopherols than that of crab apples (130.5–202.5 mg/100 g oil).

Further, Karaman et al. [113] have discussed the physicochemical, functional, and microstructural attributes of dietary fibers obtained from grape fruit, lemon, and orange seed press meals. According to their report, fiber obtained from seeds contain phytate (2–3.5 mg/g), ash (1.4–2.3%), soluble dietary fiber (5–8%), and insoluble dietary fiber (76–82%). Oil and water holding capacities of seed fibers ranged between 3.4–4.2 g/g and 4.8–7.8 g/g, respectively, with swelling capacities to be in the range of 22.4–24.6 mL/g. In a similar study, strawberry seeds obtained from industrial by-products (press cake) were investigated, which contained on average 728 g/kg of total dietary fiber [233].

## 8. Applications of Dietary Fibers in Food Products

Vast amounts of industrial wastes and by-products generated after food processing remains underutilized, leading to substantial environmental stress/pollution. Some of these can competently be used as a good source of dietary fiber; as this will not only minimize the pollution, but can also add value [234]. Numerous food and pharma products added with dietary fiber have been introduced in the market. Currently, research applications of dietary fiber remains mainly focused on dairy products, beverages, meat products, and flour products, or as food additives [97].

Some of the soluble fibers such as pectin, carboxymethyl-cellulose, inulin, and guar gum are being used as functional ingredients in dairy products such as milk, yogurt, and ice cream [97]. Numerous studies have proved that the addition of variable amounts of dietary fiber in yogurt not only improves its nutritive value, but is also able to influence its texture, rheological characteristics, consistency, and overall consumer acceptability [235]. Yogurt fortified with dietary fiber obtained from lemon and orange has exhibited good overall acceptability [236]. Staffolo et al. [237] have reported that yogurt enriched with 1.3% inulin (from wheat and bamboo) and fibers (from apple) to be a promising possibility for elevated fiber intake, which also gained increased consumer acceptability. Hashim et al. [238] fortified date fiber in different ratios (0, 1.5, 3.0, 4.5%) and wheat bran (1.5%) into fresh yogurt and studied its effects. In comparison with control yogurt, fiber fortified yogurt exhibited a significant effect on acidity, and had a darker color with firm texture. However, yogurt enriched with 3% fiber also showed similar texture firmness, smoothness, sourness, sweetness, and overall acceptability. Hence, addition of up to 3% date fiber into yogurt produced acceptable yogurt with health beneficial effects [238].

At present, various dietary fiber enriched flour products are widely available in the market. Dietary fibers are generally fortified in flours or flour based products such as biscuits, whole grain

bread, steamed bread, and noodles [97]. Partially hydrolyzed guar gum, as a source of soluble fiber has been investigated for its potential use in the production of fiber-enriched noodles imparting rich health benefits. This study showed that the addition of 1–5 g dietary fiber per 100 g of flour has substantial effects on adhesiveness, hardness, and cohesiveness of noodles [239]. A recent study revealed that noodles fortified with dietary fiber rich banana flour to have had high nutritional quality and consumer acceptability [74]. Wheat bran can be used as a dietary fiber source in the production of steamed bread and noodles, while high quality noodles are produced with the addition of 5–10% dietary fiber [240]. In a similar study, it was witnessed that addition of banana peel powder as a source of dietary fiber to have improved the quality of chapatti [241]. Combination of fiber during bread making process is reported to enhance water hydration properties of the flour. Toma et al. [242] reported bread prepared with the addition of potato peel and replaced with wheat bran to have higher water holding capacity, total dietary fiber, and essential mineral contents. Cakes prepared with the addition of 25% apple pomace and wheat flour blend possessed higher consumer acceptable ratings. Apart from other properties, fortification of apple pomace also provided a satisfying fruity flavor [86]. In a study, Nassar et al. [243] proposed 15% of orange pulp and peel to be used as an added ingredient in biscuits processing. Pulp and peel are considered to be a rich source of dietary fiber and bioactive compounds like carotenoids and flavonoids. Further, use of defatted rice bran is recommend as a substitution for wheat flour during cookies preparation. This substitution did not cause any significant effects on sensory or physical properties, but a significantly improvement in minerals, dietary fiber, and protein were recorded in the cookies [244].

Meat is often considered to be deficient in dietary fiber content; thus, attempts are being made to fortify dietary fiber from various sources into numerous meat products such as meatballs, surimi, and sausages to enhance the nutritional value. Addition of fiber in meat and/or meat products is more common nowadays, as its addition can efficiently provide longer shelf life, higher quality, as well as improve various processing characteristics [245]. Dietary fibers such as cellulose, pectin, or fiber extracted from rice, maize, wheat, and beetroot can be used to improve the texture of various meat products including salami and sausages. Meanwhile, it is also suitable to be used in the preparation of low-fat meat products, such as 'dietetic hamburgers'. Since, dietary fibers are also capable of elevating hydration properties; their inclusion in meat can contribute rich juiciness [246]. Oat fiber can be used as an appropriate fat replacement in ground pork and beef sausage based products owing to their high water retention and ability to improve texture and color [20]. Verma et al. [247] combined numerous fiber sources (pea hull flour, gram hull flour, apple pulp, and bottle gourd) to produce high fiber functional chicken nuggets with low fat and low salt. The effects of dietary fibers obtained from pineapple were also investigated for its effects on the physicochemical and textural characteristics of sausages [248]. Researchers have attempted to fortify dietary fiber into other meat products in order to improve the overall quality. Fortification of dietary fiber into meat emulsion can improve emulsion stability and viscosity and decrease cooking loss [249]; moreover, dietary fiber may also be associated with the rheological attributes of meat emulsion [250].

Addition of dietary fiber can increase the stability and viscosity in beverages and drinks. Soluble dietary fiber is often used due to its higher dispersible property in water as compare to insoluble fiber. Pectin, cellulose, and β-glucans are other soluble fibers [97] which gain potential applications. Oat fiber has been supplemented into fruit and vegetable juices, instant beverages (breakfast drinks, milk shake, sport drink, ice tea, wine), and other snack products. Dietary fibers can also be incorporated in beverages prepared for people with special needs for weight loss [251]. For example, FIBRALAX is a powdered drink prepared from pineapple peel dietary fibers, which contains 26% of dietary fiber [142]. Addition of dietary fiber as an additive to bakery-based products (as replacement of fat) can improve nutritional attributes without loss of quality [252,253]. Desserts such as frozen yogurt and ice creams contain high fat levels, which have its specific functionalities. Fortification of fiber ingredients such as cellulose gels, guar gums, and alginates can be a good substitute of fat, which is also envisaged to

improve emulsion, viscosity, and foam, reduce syneresis, control melting properties, and stimulate formation of ice crystals [254].

With the ongoing research activities on dietary fiber applications, it is worthwhile to further explore fruits and vegetable wastes and by-products for their potential applications in food and pharma based products.

## 9. Conclusions

This review paper highlights that ample amounts of non-edible and edible parts of fruits and vegetables are wasted along the entire agri-food supply chain. The wastes generated are attributed to lack of appropriate pre- and post-harvest handling and processing operations. However, available literature and databases indicate that these wastes and/or by-products are rich in bioactive compounds including dietary fibers, which possess high potential usage in food and pharmaceutical industries. Studies have clearly indicated increased utilization of fiber supplementation to be beneficial for human health. By use of various fruit and vegetable wastes and/or by-products, currently in the global market, a wide range of dietary fiber supplements is designed. Future studies are warranted, wherein green technologies such as water, ethanol, and/or steam extraction, supercritical $CO_2$, and others can be adopted for extraction of dietary fiber from wastes and by-products. Besides, use of dietary fiber extracted from vegetal wastes can also find wide applications as a value-added ingredient while developing novel healthy food products. Additionally, research on extraction of dietary fiber from underutilized fruits and vegetables (e.g., wild berries, wild mushroom, wild crucifers, etc.) remains in infancy stage. However, a wide gap still exists to exploit the potential applications of dietary fiber in food and pharmaceutical industries in a sustainable manner to develop value added healthy products, for development of functional foods, and for bio-fortification/supplementation in various types of foods and beverages. Opportunities also exist to explore dietary fiber from wastes and by-products to develop livestock feed. These gaps are expected to be filled in the near future for sustainable utilization food industrial wastes and by-products.

**Author Contributions:** S.H. and R.B. have contributed directly in writing, planning, and editing of this review article; I.J. was involved in project administration. All authors have read and agreed to the published version of the manuscript.

**Funding:** The theme of this review article is based on our ongoing project—VALORTECH, which has received funding from the European Union's Horizon 2020 research and innovation program under grant agreement No 810630.

**Conflicts of Interest:** The authors declare no conflict of interest in this review.

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
