# Peer review of "Dietary Fiber from Underutilized Plant Resources—A Positive Approach for Valorization of Fruit and Vegetable Wastes"

_sustainability, doi:10.3390/su12135401_

Round 1

Reviewer 1 Report

The m/s by Hussain et al. provides a well-structured overview of dietary fibre content in fruit and vegetable by-products.

The review reads well and has a logical flow. However, although there is discussion around extraction methodologies in a few sections (and featuring in Figure 1), there is lack of critical evaluation of the most promising extraction technologies for dietary fibre. This element should be emphasised more in the article, as it is often the bottleneck for larger scale implementation of the given technology. 

Figure 3 seems a bit out of place, as it is very briefly mentioned in this review, however it constitutes a great deal of research topic on its own. Authors should either consider expanding more on the topic (carbohydrate structure-function relationship, type of carbohydrate and fermentability etc) or omit Figure 3.

Author Response

Please see attached response file

Reviewer 2 Report

GENERAL COMMENTS

The article adequately organizes and analyzes a wealth of existing information about the fiber content in agri-food wastes. I think the work will be useful in the scientific field and should be published after a major revision.

The objective is not adequately defined neither in the abstract nor in the manuscript text. Lines 20-23: Authors say: “Based on this the review has been designed to support ….for extraction of health promoting dietary fibers”. I think it is neither accurate nor correct and it should be precisely redefined.

Lines 107-111: The author defines more precisely the aim of the work that he had already stated inaccurately in lines 55-59. I think the goal has to be precisely defined just once at the end of the introduction.

Throughout the manuscript, the following ideas are repeated:

The sustainability of the food system requires the waste valorisation for a circular economy.

Fruit and vegetable wastes are rich in fiber and bioactive components with a positive effect on disease prevention.

The residues can be used to obtain ingredients of interest to the pharmaceutical and food industries.

Please, unify ideas, structure the sections well and avoid repetition.

Sections 2 and 7 can be merged into a single section 2.

The author uses the adjective "vital" recurrently to refer to something that is important or necessary. Please use the language more rigorously and substitute more appropriate adjectives.

SPECIFIC COMMENTS

Line 13: Change “Food industries” by “Agri-food industries”.

Line 14 and 34: Change “research arena” by research area”.

Line 49-51: a reference is needed.

Lines 68 and 69: It should be removed. The sentence is unnecessary.

Lines 91-93: The idea is discussed in depth in the next section. It should be removed here.

Lines 93-96: The author should also cite the non-thermal technologies that are used for extraction or facilitate extraction: HPH, PEF, US.

Lines 100-101: Repeat lines 81-82.

Figure 1:

- When the authors cite Mucilage as a soluble fiber, they refers Food additives as the source. Please change by a correct vegetable source.

- Non-thermal processing treatments (HPH, PEF and US) should be included into Extraction methods.

Lines 186-189: waste concept has been previously defined by EU. Please, include de reference.

Line 190: Change “very” by “vary”???

Line 192- 193: A reference is required.

Line 198: Repeat lines 189-190.

Line 208-211: Repetition.

Figure 2: Functional food production belongs to Food sector and not to the Pharmaceutical sector.

Line 239: Please change the verb “opined”.

Line 259-260: A reference is required.

Line 264: delete the statment repeated: “blackcurrant, red currant, gooseberry, rowanberry and chokeberry, respectively”

Lines 267-287: The values given in the table do not need to be repeated in the text. Use the text to comment on other aspects such as extraction procedure, soluble and insoluble fiber ratio ...

Line 290: delete “along”.

Lines 289-311: The values given in the table do not need to be repeated in the text. Use the text to comment on other aspects such as extraction procedure, soluble and insoluble fiber ratio ...

Lines 350-351: what about its water and oil intaraction properties? are these properties related with soluble and insoluble fiber proportion?

Lines 432-433: The same reference is included twice.

Line 475: left over parenthesis.

Line 688: nowadays.

Lines 737- 739: The idea is repeated. It should be omitted.

Reviewer 3 Report

The manuscript entitled "Valorization of Fruit and Vegetable Wastes to Obtain Bioactive Dietary Fibers" has high scientific relevance. The work is well written and the discussion is well-founded. The table is appropriate to represent the results. However, I would like to encourage the authors, to let know the readers, that the topic of valorization of fruit and vegetable by-products as a source of bioactive dietary fibers, is still "open", especially in the case of seeds. For instance, the apple seeds were considered, and are well studied, as a source of oil, lipophilic compounds (sterols, tocopherols, tocotrienols, squalene) and hydrophilic (phenolic molecules), however, the knowledge about the bioactive dietary fibers in the seeds is still quite poor or its lack. Therefore, I would suggest one more short paragraph at the end of the manuscript, linked to this issue, to highlight that it is still a lot what to do in the topic of valorization of fruit and vegetable wastes as bioactive dietary fibers. The papers listed below can be helpful.

"Seeds recovered from industry by-products of nine fruit species with a high potential utility as a source of unconventional oil for biodiesel and cosmetic and pharmaceutical sectors. Industrial Crops and Products. DOI: 10.1016/j.indcrop.2016.01.021";

"Unique variability of tocopherol composition in various seed oils recovered from by-products of apple industry: Rapid and simple determination of all four homologues (α, β, γ and δ) by RP-HPLC/FLD. Food Chemistry, DOI: 10.1016/j.foodchem.2014.09.051";

"Lipophilic composition of eleven apple seed oils: A promising source of unconventional oil from industry by-products. Industrial Crops and Products, DOI: 10.1016/j.indcrop.2014.06.003";

"Seed oils recovered from industrial fruit by‐products are a rich source of tocopherols and tocotrienols: Rapid separation of α/β/γ/δ homologues by RP‐HPLC/FLD. European Journal of Lipid Science and Technology, DOI: 10.1002/ejlt.201400566";

"Phenolic compounds in different fruit parts of crab apple: Dihydrochalcones as promising quality markers of industrial apple pomace by-products. Industrial Crops and Products, DOI: 10.1016/j.indcrop.2015.05.030".

Round 2

Reviewer 2 Report

The changes suggested have been answered succesfully.